# Long-term outcomes of peripheral arterial disease patients with significant coronary artery disease undergoing percutaneous coronary intervention

**Byoung Geol Choi[1]☯\*, Ji-Yeon Hong[2]☯, Seung-Woon Rha[3]\*, Cheol Ung Choi[3], Michael S. Lee[4]**

1 Cardiovascular Research Institute, Korea University, Seoul, Korea, 2 Division of Cardiology, Nowon Eulji Hospital, Eulji University, Seoul, Korea, 3 Cardiovascular Center, Korea University Guro Hospital, Seoul, Korea, 4 UCLA Medical Center, Los Angeles, California, United States of America

☯ These authors contributed equally to this work.

\* trv940@korea.ac.kr (BGC); swrha617@yahoo.co.kr (S-WR)

## Abstract

### Background

Patients with peripheral arterial disease (PAD) have known to a high risk of cardiac mortality. However, the effectiveness of the routine evaluation of coronary arteries such as routine coronary angiography (CAG) in PAD patients receiving percutaneous transluminal angioplasty (PTA) is unclear.

### Methods

A total of 765 consecutive PAD patients underwent successful PTA and 674 patients (88.1%) underwent routine CAG. Coronary artery disease (CAD) was defined as angiographic stenosis ≥70%. Patients were divided into three groups; 1) routine CAG and a presence of CAD (n = 413 patients), 2) routine CAG and no CAD group (n = 261 patients), and 3) no CAG group (n = 91 patients). To adjust for any potential confounders that could cause bias, multivariable Cox-proportional hazards regression and propensity score matching (PSM) analysis was performed. Clinical outcomes were evaluated by Kaplan-Meier curved analysis at 5-year follow-up.

### Results

In this study, the 5-year survival rate of patients with PAD who underwent PTA was 88.5%. Survival rates were similar among the CAD group, the no CAD group, and the no CAG group, respectively (87.7% vs. 90.4% vs. 86.8% P = 0.241). After PSM analysis between the CAD group and the no CAD group, during the 5-year clinical follow-up, there were no differences in the incidence of death, myocardial infarction, strokes, peripheral revascularization, or target extremity surgeries between the two groups except for repeat PCI, which was higher in the CAD group than the non-CAD group (9.3% vs. 0.8%, P<0.001).

**Data Availability Statement:** All relevant data are within the paper.

**Funding:** This work was supported by the Korea Medical Device Development Fund grant funded by the Korea government (the Ministry of Science and

ICT, the Ministry of Trade, Industry and Energy, the Ministry of Health & Welfare, the Ministry of Food and Drug Safety) (Project Number: 9991006707, KMDF_PR_20200901_0034).

**Competing interests:** The authors have declared that no competing interests exist.

## Conclusion

PAD patients with CAD were expected to have very poor long-term survival, but they are shown no different long-term prognosis such as mortality compared to PAD patients without CAD. These PAD patients with CAD had received PCI and/or optimal medication treatment after the CAG. Therefore a strategy of routine CAG and subsequent PCI, if required, appears to be a reasonable strategy for mortality risk reduction of PAD patients. Our results highlight the importance for evaluation for CAD in patients with PAD.

## Introduction

Peripheral arterial disease (PAD) reduces the quality of life and is associated with poor long-term clinical prognosis [1,2]. Also, coronary artery disease (CAD) is a leading cause of death in many geographies of the world [3]. It shares many risk factors with PAD including age, smoking, diabetes, and chronic kidney disease [2–4]. Because atherosclerosis is a systemic process that can affect multiple vascular territories, patients with either PAD or CAD commonly have the other condition [1,2,5,6]. The severe CAD has been observed in 54% to 69% of patients with PAD [7–10]. Patients with PAD have particularly a high mortality rate from cardiovascular events [6,11–13]. Thus, patients with both PAD and CAD are expected to have a particularly poor long-term prognosis.

Percutaneous transluminal angioplasty (PTA) is commonly used to improve claudication of the extremities and treat critical limb ischemia [7,14]. The use of cardio-protective drugs such as antiplatelet therapy, statin, and angiotensin-converting enzyme inhibitors in PAD helps to improve survival, but the introduction of these guidelines is more than a decade behind CAD [15]. Therefore, evaluating the presence of CAD, which can lead to coronary revascularization and optimal medical treatment, can improve both short- and long-term surviaval in patients receiving PTA [8,16,17]. However, the effective of routine evaluation of coronary arteries in patients receiving PTA is unclear [11,18]. In the present study, we evaluated the long-term clinical outcomes of the strategy of routine coronary angiography (CAG) and subsequent percutaneous coronary intervention (PCI) in patients with PAD who underwent PTA.

## Methods

We obtained data from PTA registry of Korea University Guro Hospital (KUGH), Seoul, South Korea. This registry has been described in detail in previous studies [8,19,20]. In brief, this is a single-center, prospective, all-comers registry which started in 2006 and was designed to reflect "real world" practice. Data are collected by trained study-coordinators using standardized case report forms.

### Ethical approval

Participants or their legal guardians were given a thorough literal and verbal explanation of the study procedures before asking for written consent to participate in the study. The study protocol was approved by the Medical Device Institutional Review Board (MD-IRB) of KUGH (IRB protocol #MD12018).

## Data source and population

A total of 765 consecutive PAD patients underwent successful PTA and 674 patients underwent routine CAG. For the remaining 91 patients who did not receive the CAG, but the cardiac function was evaluated non-invasively by the cardiologist. CAD was defined as angiographic stenosis ≥70% in the main epicardial coronary vessels. Patients were divided into three groups; 1) routine CAG and a presence of CAD (the CAD group: 413 patients), 2) routine CAG without CAD (the no CAD group: 261 patients), and 3) no CAG group (91 patients).

## Percutaneous transluminal angioplasty

Standard techniques were used for PTA. For below-the-knee lesions, a 5 Fr Heartrail guiding catheter (Terumo) was used, and a 0.014″ guidewire was used to traverse the lesions. If intraluminal wiring failed, sub-intimal angioplasty or retrograde approach was performed. After guidewire crossing, prolonged balloon inflation with a balloon ranging in size from 1.5–3.0 mm was used for infra-popliteal vessels and 4–7 mm for femoro-popliteal vessels. Provisional stenting was performed using self-expanding nitinol stents (Xpert, Abbott Vascular, or Maris Deep; Medtronic-Invatec) if balloon angioplasty results were suboptimal. For the ilio-femoral lesion, 6 or 7 F guiding sheath was placed and true lumen angioplasty was attempted to treat chronic total occlusion (CTO) of the superficial femoral artery (SFA) and iliac artery using dedicated 0.018″ CTO wires. If the intraluminal approach was unsuccessful, subintimal angioplasty using 0.035″ soft Terumo wire (1.5 J curve) with a 5 Fr angiocatheter support was performed for longer CTO lesions with prolonged balloon inflation with adequate size. If balloon angioplasty results were favorable, PTA with drug-coated balloons (DCB) was performed. If the balloon angioplasty result was suboptimal, provisional stenting with self-expanding nitinol stents or drug-eluting stents was performed. Wiring of the true lumen was performed for shorter CTO lesions. Re-entry with CTO wires or a re-entry device (Outback catheter, Cordis) was used if the subintimal wiring failed to re-enter the distal true lumen for femoropopliteal CTO lesions. A retrograde approach from the distal SFA, popliteal, or infra-popliteal arteries was used in selected cases.

## Study definitions [21,22]

PAD was defined as ischemic pain at rest, an ulcer, or gangrene in one or both legs attributed to objectively proven arterial occlusive disease. The main epicardial coronary arteries were defined as having a reference vessel diameter of >2.5 mm at left main-, left anterior descending-, left circumflex-, right coronary-, and the ramus artery. Major adverse cardiovascular and cerebrovascular events (MACCE) were defined as the composite of total death, myocardial infarction (MI), stroke, and revascularization including PCI and coronary artery bypass graft (CABG). Target lesion revascularization (TLR) was defined as ischemia-induced PTA of the target lesion due to restenosis or re-occlusion within the balloon angioplasty site, stent, or in the adjacent 5 mm of the distal or proximal segment. Target extremity revascularization (TER) was defined as clinically-driven PTA of the target lesion or any segment of the same limb containing the target lesion. Major adverse limb events (MALE) were defined as the composite of TER and target extremity surgery.

## Statistical analysis

Differences in continuous variables, among the three groups, were evaluated by ANOVA or Kruskal-Wallis, and post-hoc analysis between the two groups were evaluated by Hochberg or

Dunnett-T3 test. Differences in continuous variables between the two groups were evaluated using the unpaired t-test or Mann-Whitney rank test. Data are expressed as means ± standard deviations. For discrete variables, differences are expressed as counts and percentages and were analyzed with the χ2 test or Fisher's exact test. Multivariable Cox-proportional hazards regression, which includes baseline-confounding factors, was used for assessing independent impact factors. To adjust for any potential confounders, propensity score matching (PSM) analysis was performed using a logistic regression model. Matching was performed with a 1:1 matching protocol using the nearest neighbor matching algorithm with a caliper width less than 0.01 the standard deviation of the propensity score. Clinical outcomes that occurred over a period of 5-year were analyzed by Kaplan-Meier analysis, and differences between groups were compared with the log-rank test before and after PSM. For all analyses, a two-sided p < 0.05 was considered statistically significant. All data were analyzed using SPSS (version 20.0, SPSS-PC, Inc. Chicago, Illinois).

### Study endpoints

Primary endpoints were MACCE as the composite of total death, MI, stroke, revascularization such as PCI and CABG at 5-year follow-up. Secondary endpoints were TLR, TER, and target extremity surgery after PTA at 5-year clinical follow-up. Patients were followed up at one month and then every 6 months after the PTA procedure as well as whenever cardiovascular ischemic symptoms occurred. Follow-up was performed with face-to-face interviews at the regular outpatient clinic, medical chart reviews, and/or telephone contact.

### Results

A total of 765 consecutive PAD patients underwent successful PTA and 674 patients subsequently underwent a routine CAG. The baselines clinical characteristics were similar between the CAG group (the CAD group and the no CAD group) and the no CAG group. In the CAG group, a total of 61.2% patients (413/674) were diagnosed with severe CAD in the main epicardial coronary arteries. Among the patients with CAD, 15.0% (62/413) and 59.0% (244/413) had left main disease and multi-vessel disease, respectively. Among the CAD patients, 71.6% (296/413) treated CAD lesions by PCI and/or CABG, during or after the PTA admission period based upon the physician's discretion. The CAD group had more elderly patients, diabetics, and higher levels of creatinine than patients in the no CAD group (Table 1). The no CAD group had a higher prevalence of Buerger's disease and were more likely to be current smokers and alcohol drinkers than the CAD group. The CAD group had a higher prevalence of femoral lesions (49.9% vs. 40.8%, p = 0.011) (Table 2).

Procedural and in-hospital complications after PTA were similar among the three groups (Table 3). The CAD group received more clopidogrel, sarpogrelate, angiotensin-converting enzyme inhibitors, and angiotensin receptor blockers (ARB), β-blocker than the non-CAD, and no-CAG group (Table 3).

In this study, the 5-year survival rate of patients with PAD who underwent PTA was 88.5%. Survival rates were similar among the CAD group, the no CAD group and the no CAG group up to 5-year of clinical follow-up, respectively (87.7% vs. 90.4% vs. 86.8% P = 0.241) (Table 4, Fig 1). There was a trend toward a higher incidence of cardiac death in the no CAG group (3.0% vs. 0.8% vs. 3.6%, P = 0.066) than the CAD group and the no CAD group. Coronary revascularization was higher in the CAD group (9.4% vs. 0.8% vs. 4.9% P = 0.005). There was no difference between the two groups in PTA-related events such as peripheral revascularization or target extremity surgery (Table 4).

**Table 1. Baseline characteristics of the entire cohort and propensity-matched groups.**

| | All patients | | | | Matched patients | | | |
|---|---|---|---|---|---|---|---|---|
| | CAG | | No CAG | | CAG | | | |
| Variable, N (%) | CAD (n = 413 Pts) | No CAD (n = 261 Pts) | (n = 91 Pts) | P value | CAD (n = 160 Pts) | Non-CAD (n = 160 Pts) | P value | S.diff |
| Sex, male | 314 (76.0) | 204 (78.1) | 78 (85.7) | 0.130 | 125 (78.1) | 123 (76.8) | 0.789 | -0.14 |
| Age, years | 69.1 ± 9.1 | 66.3 ± 12.5 | 67.5 ± 10.1 | 0.003 | 67.9 ± 9.6 | 68.8 ± 11.1 | 0.210 | -0.08 |
| Body mass index, kg/m$^2$ | 23.1 ± 3.3 | 23.3 ± 3.2 | 22.9 ± 3.2 | 0.497 | 23.4 ± 3.2 | 23.4 ± 3.1 | 0.483 | 0.02 |
| **Final diagnosis** | | | | | | | | |
| Diabetic foot ulcer | 239 (57.8) | 139 (53.2) | 59 (64.8) | 0.143 | 96 (60.0) | 94 (58.7) | 0.820 | -0.16 |
| Wound | 258 (62.4) | 166 (63.6) | 62 (68.1) | 0.597 | 105 (65.6) | 103 (64.3) | 0.815 | -0.16 |
| Gangrene | 138 (33.4) | 88 (33.7) | 30 (32.9) | 0.991 | 57 (35.6) | 47 (29.3) | 0.233 | -1.10 |
| Claudication | 73 (17.6) | 56 (21.4) | 21 (23.0) | 0.327 | 31 (19.3) | 32 (20.0) | 0.888 | 0.14 |
| Resting pain | 44 (10.6) | 34 (13.0) | 11 (12.0) | 0.639 | 21 (13.1) | 19 (11.8) | 0.735 | -0.35 |
| Buerger's disease | 6 (1.4) | 15 (5.7) | 4 (4.3) | 0.008 | 3 (1.8) | 2 (1.2) | > 0.99 | -0.50 |
| Other | 37 (8.9) | 7 (2.6) | 0 (0.0) | <0.001 | 5 (3.1) | 6 (3.7) | 0.759 | 0.34 |
| **Risk Factors** | | | | | | | | |
| Hypertension | 302 (73.1) | 178 (68.1) | 61 (67.0) | 0.279 | 113 (70.6) | 117 (73.1) | 0.619 | 0.30 |
| Diabetes mellitus | 317 (76.7) | 169 (64.7) | 74 (81.3) | <0.001 | 122 (76.2) | 115 (71.8) | 0.372 | -0.51 |
| Insulin | 134 (32.4) | 75 (28.7) | 34 (37.3) | 0.285 | 62 (38.7) | 51 (31.8) | 0.198 | -1.16 |
| Oral medication | 127 (30.7) | 70 (26.8) | 35 (38.4) | 0.111 | 36 (22.5) | 49 (30.6) | 0.100 | 1.58 |
| Dyslipidemia | 50 (12.1) | 34 (13.0) | 10 (10.9) | 0.866 | 19 (11.8) | 16 (10.0) | 0.591 | -0.57 |
| Strokes | 90 (21.7) | 44 (16.8) | 19 (20.8) | 0.289 | 31 (19.3) | 33 (20.6) | 0.780 | 0.28 |
| Hemorrhagic | 9 (2.1) | 8 (3.0) | 4 (4.3) | 0.467 | 0 (0.0) | 4 (2.5) | 0.123 | 2.24 |
| Ischemic | 81 (19.6) | 36 (13.7) | 16 (17.5) | 0.152 | 31 (19.3) | 29 (18.1) | 0.775 | -0.29 |
| Chronic renal insufficiency | 132 (31.9) | 67 (25.6) | 32 (35.1) | 0.122 | 56 (35.0) | 52 (32.5) | 0.636 | -0.43 |
| Dialysis | 82 (19.8) | 40 (15.3) | 19 (20.8) | 0.273 | 32 (20.0) | 30 (18.7) | 0.777 | -0.28 |
| Congestive heart failure | 29 (7.0) | 14 (5.3) | 3 (3.2) | 0.345 | 11 (6.8) | 12 (7.5) | 0.829 | 0.23 |
| History of smoking | 213 (51.5) | 148 (56.7) | 52 (57.1) | 0.348 | 81 (50.6) | 83 (51.8) | 0.823 | 0.18 |
| Current of smoking | 118 (28.5) | 108 (41.3) | 31 (34.0) | 0.003 | 46 (28.7) | 48 (30.0) | 0.806 | 0.23 |
| History of alcohol drinking | 131 (31.7) | 107 (40.9) | 33 (36.2) | 0.049 | 52 (32.5) | 55 (34.3) | 0.722 | 0.32 |
| Currently alcohol drinking | 80 (19.3) | 76 (29.1) | 18 (19.7) | 0.010 | 29 (18.1) | 32 (20.0) | 0.669 | 0.43 |
| **Laboratory findings** | | | | | | | | |
| Hemoglobin, mg/dL | 11.3 ± 1.9 | 11.9 ± 2.1 | 11.3 ± 1.9 | 0.004 | 11.5 ± 2.0 | 11.6 ± 2.1 | 0.548 | -0.07 |
| Fasting glucose, mg/dL | 151 ± 74 | 137 ± 77 | 145 ± 77 | 0.074 | 154 ± 70 | 144 ± 80 | 0.068 | 0.13 |
| Hemoglobin A1c, % | 7.2 ± 1.6 | 6.8 ± 1.5 | 7.1 ± 1.4 | 0.015 | 7.3 ± 1.7 | 7.0 ± 1.4 | 0.278 | 0.18 |
| Total cholesterol, mg/dL | 146 ± 42 | 147 ± 46 | 137 ± 34 | 0.235 | 146 ± 47 | 147 ± 47 | 0.645 | -0.04 |
| Triglycerides, mg/dL | 126 ± 98 | 129 ± 89 | 117 ± 59 | 0.631 | 124 ± 70 | 135 ± 104 | 0.711 | -0.13 |
| HDL cholesterol, mg/dL | 37 ± 12 | 38 ± 13 | 35 ± 11 | 0.108 | 36 ± 13 | 37 ± 13 | 0.412 | -0.11 |
| LDL cholesterol, mg/dL | 88.6 ± 34.9 | 90.9 ± 36.9 | 79.8 ± 31.8 | 0.065 | 88 ± 37 | 89 ± 37 | 0.779 | -0.03 |
| hs-CRP, mg/L | 23.1 ± 47.5 | 18.3 ± 37.3 | 22.6 ± 32.7 | 0.690 | 22.5 ± 45 | 20 ± 35.7 | 0.442 | 0.06 |
| Creatinine, mg/dL | 2.2 ± 2.8 | 1.7 ± 2.2 | 2.4 ± 2.9 | 0.029 | 2.3 ± 2.8 | 2.0 ± 2.5 | 0.200 | 0.11 |

Data are presented as N (%) or mean ± standard deviation. CAD, coronary artery disease; S.diff, standardized difference; HDL, high-density lipoprotein; LDL, low-density lipoprotein; hs-CRP, high-sensitive C-reactive protein.

PSM analysis between the CAD group and the non-CAD group yielded two matched groups (160 pairs, n = 320) with balanced baseline characteristics (Tables 1–3). There were no differences in MI, strokes, peripheral revascularization, or target extremity surgeries between the two groups over the 5-year clinical follow-up period except for non-cardiac death and

**Table 2. Coronary angiographic and clinical limb characteristics.**

| | All patients | | | | Matched patients | | | |
| --- | --- | --- | --- | --- | --- | --- | --- | --- |
| | CAG | | No CAG | | CAG | | | |
| Variable, N (%) | CAD (n = 413 Pts) (n = 545 Limb) | No CAD (n = 261 Pts) (n = 313 Limb) | (n = 91 Pts) (n = 110 Limb) | P value | CAD (n = 160 Pts) (n = 197 Limb) | Non-CAD (n = 160 Pts) (n = 201 Limb) | P value | S.diff |
| **Coronary artery information (No. patients)** | | | | | | | | |
| Treated coronary artery disease | 296 (71.6) | 0 (0.0) | 22 (24.1) | <0.001 | 113 (70.6) | 0 (0.0) | < 0.001 | -11.93 |
| CABG | 25 (6.0) | 0 (0.0) | 7 (7.6) | <0.001 | 12 (7.5) | 0 (0.0) | < 0.001 | -3.87 |
| PCI | 283 (68.5) | 0 (0.0) | 17 (18.6) | <0.001 | 107 (66.8) | 0 (0.0) | < 0.001 | -11.60 |
| PCI in PTA | 214 (51.8) | 0 (0.0) | 0 (0.0) | <0.001 | 84 (52.5) | 0 (0.0) | < 0.001 | -10.27 |
| Coronary artery | 244 (59.0) | 0 (0.0) | | <0.001 | | | | |
| Left main | 62 (15.0) | 0 (0.0) | | <0.001 | 20 (12.5) | 0 (0.0) | < 0.001 | -5.00 |
| Left anterior descending | 251 (60.7) | 0 (0.0) | | <0.001 | 91 (56.8) | 0 (0.0) | < 0.001 | -10.70 |
| Left circumflex | 233 (56.4) | 0 (0.0) | | <0.001 | 93 (58.1) | 0 (0.0) | < 0.001 | -10.81 |
| Right coronary artery | 236 (57.1) | 0 (0.0) | | <0.001 | 91 (56.8) | 0 (0.0) | < 0.001 | -10.70 |
| Disease artery, N | 1.8 ± 0.8 | 0.0 ± 0.0 | | <0.001 | 1.8 ± 0.7 | - | < 0.001 | 3.26 |
| **Peripheral artery information, limbs** | | | | | | | | |
| Ankle brachial index | 0.67 ± 0.37 | 0.72 ± 0.38 | 0.83 ± 0.30 | 0.011 | 0.67 ± 0.40 | 0.66 ± 0.42 | 0.863 | 0.03 |
| Limb site | | | | | | | 0.619 | -0.36 |
| Right | 289 (53.0) | 149 (47.6) | 56 (50.9) | 0.310 | 99 (50.2) | 96 (47.7) | | |
| Left | 256 (46.9) | 164 (52.3) | 54 (49.0) | 0.310 | 98 (49.7) | 105 (52.2) | | |
| **Rutherford grade, limbs** | | | | | | | 0.74 | 1.00 |
| Grade 0 (Category 0) | 51 (9.3) | 16 (5.1) | 3 (2.7) | <0.001 | 9 (4.5) | 14 (6.9) | | |
| Grade 1 | 147 (26.9) | 98 (31.3) | 25 (22.7) | | 56 (28.4) | 56 (27.8) | | |
| Category 1 | 29 (5.3) | 17 (5.4) | 2 (1.8) | | 11 (5.5) | 11 (5.4) | | |
| Category 2 | 42 (7.7) | 23 (7.3) | 7 (6.3) | | 19 (9.6) | 14 (6.9) | | |
| Category 3 | 76 (13.9) | 58 (18.5) | 16 (14.5) | | 26 (13.1) | 31 (15.4) | | |
| Grade 2 | 180 (33) | 114 (36.4) | 57 (51.8) | | 77 (39.0) | 80 (39.8) | | |
| Category 4 | 46 (8.4) | 38 (12.1) | 14 (12.7) | | 14 (7.1) | 25 (12.4) | | |
| Category 5 | 134 (24.5) | 76 (24.2) | 43 (39) | | 63 (31.9) | 55 (27.3) | | |
| Grade 3 (Category 6) | 167 (30.6) | 85 (27.1) | 25 (22.7) | | 55 (27.9) | 51 (25.3) | | |
| **Location, limbs** | | | | | | | | |
| Distal aorta | 16 (2.9) | 14 (4.4) | 4 (3.6) | 0.499 | 11 (5.5) | 9 (4.4) | 0.614 | -0.49 |
| Iliac artery | 164 (30.0) | 81 (25.8) | 21 (19.0) | 0.046 | 59 (29.9) | 52 (25.8) | 0.364 | -0.77 |
| Femoral artery | 272 (49.9) | 128 (40.8) | 44 (40.0) | 0.016 | 96 (48.7) | 87 (43.2) | 0.276 | -0.80 |
| Popliteal artery | 43 (7.8) | 35 (11.1) | 19 (17.2) | 0.008 | 22 (11.1) | 14 (6.9) | 0.144 | -1.40 |
| Anterior tibia artery | 254 (46.6) | 156 (49.8) | 65 (59.0) | 0.055 | 98 (49.7) | 106 (52.7) | 0.551 | 0.42 |
| Posterior tibia artery | 210 (38.5) | 111 (35.4) | 45 (40.9) | 0.521 | 69 (35.0) | 72 (35.8) | 0.868 | 0.13 |

(*Continued*)

**Table 2.** (Continued)

| Variable, N (%) | All patients CAG CAD (n = 413 Pts) (n = 545 Limb) | No CAD (n = 261 Pts) (n = 313 Limb) | No CAG (n = 91 Pts) (n = 110 Limb) | P value | Matched patients CAG CAD (n = 160 Pts) (n = 197 Limb) | Non-CAD (n = 160 Pts) (n = 201 Limb) | P value | S.diff |
|---|---|---|---|---|---|---|---|---|
| Peroneal artery | 105 (19.2) | 56 (17.8) | 26 (23.6) | 0.422 | 40 (20.3) | 41 (20.3) | 0.982 | 0.02 |

Data are presented as N (%) or mean ± standard deviation. CAD, coronary artery disease; S.diff, standardized difference; CABG, coronary artery bypass graft surgery; PCI, percutaneous coronary intervention; PTA, percutaneous transluminal angioplasty.

**Table 3. Post-procedural complications and medications.**

| Variable, N (%) | All patients CAG CAD (n = 413 Pts) | No CAD (n = 261 Pts) | No CAG (n = 91 Pts) | P value | Matched patients CAG CAD (n = 160 Pts) | Non-CAD (n = 160 Pts) | P value |
|---|---|---|---|---|---|---|---|
| **Complications at access site** | 51 (12.3) | 24 (9.1) | 6 (6.5) | 0.181 | 10 (6.2) | 16 (10.0) | 0.22 |
| Arteriovenous fistula | 3 (0.7) | 2 (0.7) | 1 (1.0) | 0.851 | 1 (0.6) | 2 (1.2) | > 0.99 |
| Pseudo-aneurysm | 4 (0.9) | 4 (1.5) | 1 (1.0) | 0.890 | 0 (0.0) | 3 (1.8) | 0.248 |
| Hematoma | 48 (11.6) | 20 (7.6) | 5 (5.4) | 0.088 | 10 (6.2) | 12 (7.5) | 0.659 |
| minor, < 4 cm | 28 (6.7) | 13 (4.9) | 3 (3.2) | 0.349 | 6 (3.7) | 7 (4.3) | 0.777 |
| Major, > 4 cm | 20 (4.8) | 7 (2.6) | 2 (2.1) | 0.251 | 4 (2.5) | 5 (3.1) | > 0.99 |
| **Bleeding complications** | 98 (23.7) | 42 (16.0) | 19 (20.8) | 0.059 | | | |
| Major bleeding | 9 (2.1) | 4 (1.5) | 2 (2.1) | 0.828 | 3 (1.8) | 2 (1.2) | > 0.99 |
| Gastrointestinal bleeding | 5 (1.2) | 2 (0.7) | 2 (2.1) | 0.494 | 1 (0.6) | 0 (0.0) | > 0.99 |
| Retroperitoneal bleeding | 4 (0.9) | 2 (0.7) | 0 (0.0) | >0.999 | 2 (1.2) | 2 (1.2) | > 0.99 |
| Transfusion | 172 (41.6) | 72 (27.5) | 31 (34.0) | 0.001 | 62 (38.7) | 58 (36.2) | 0.605 |
| Transfusion, pints | 3.1 ± 7.3 | 2.2 ± 6.8 | 2.0 ± 3.9 | 0.165 | 2.6 ± 6.1 | 2.9 ± 7.8 | 0.746 |
| **In-hospital complications** | | | | | | | |
| Acute limb ischemia | 5 (1.2) | 7 (2.6) | 0 (0.0) | 0.178 | 1 (0.6) | 3 (1.8) | 0.623 |
| Acute renal failure | 9 (2.1) | 3 (1.1) | 3 (3.2) | 0.398 | 3 (1.8) | 3 (1.8) | > 0.99 |
| Congestive heart failure | 3 (0.7) | 0 (0.0) | 1 (1.0) | 0.179 | - | - | - |
| Strokes | 1 (0.2) | 3 (1.1) | 0 (0.0) | 0.348 | 0 (0.0) | 2 (1.2) | 0.498 |
| Hemorrhagic | 0 (0.0) | 1 (0.3) | 0 (0.0) | 0.460 | 0 (0.0) | 1 (0.6) | > 0.99 |
| Ischemic | 1 (0.2) | 2 (0.7) | 0 (0.0) | 0.701 | 0 (0.0) | 1 (0.6) | > 0.99 |
| **Post-procedural medications** | | | | | | | |
| Aspirin | 405 (98.0) | 254 (97.3) | 88 (96.7) | 0.675 | 155 (96.8) | 155 (96.8) | > 0.99 |
| Clopidogrel | 379 (91.7) | 224 (85.8) | 80 (87.9) | 0.047 | 142 (88.7) | 142 (88.7) | > 0.99 |
| Cilostazol | 179 (43.3) | 118 (45.2) | 33 (36.2) | 0.330 | 62 (38.7) | 69 (43.1) | 0.426 |
| Anplag | 100 (24.2) | 51 (19.5) | 11 (12.0) | 0.027 | 25 (15.6) | 33 (20.6) | 0.246 |
| ARBs | 176 (42.6) | 106 (40.6) | 32 (35.1) | 0.419 | 72 (45.0) | 65 (40.6) | 0.429 |
| ACEIs | 86 (20.8) | 19 (7.2) | 10 (10.9) | <0.001 | 15 (9.3) | 16 (10.0) | 0.85 |
| Calcium channel blocker | 182 (44.0) | 106 (40.6) | 35 (38.4) | 0.501 | 60 (37.5) | 69 (43.1) | 0.305 |
| β-blocker | 143 (34.6) | 54 (20.6) | 16 (17.5) | <0.001 | 38 (23.7) | 41 (25.6) | 0.697 |
| Diuretic | 116 (28.0) | 58 (22.2) | 22 (24.1) | 0.223 | 39 (24.3) | 39 (24.3) | > 0.99 |
| Statin | 341 (82.5) | 223 (85.4) | 73 (80.2) | 0.441 | 134 (83.7) | 134 (83.7) | > 0.99 |

Data are presented as N (%) or mean ± standard deviation. CAD, coronary artery disease; S.diff, standardized difference; ARBs: Angiotensin II receptor blockers; ACEIs, angiotensin-converting enzyme inhibitors.

**Table 4. Kaplan–Meier curve analysis of clinical outcomes and log-rank test results.**

| | All patients | | | | Matched patients | | |
|---|---|---|---|---|---|---|---|
| | CAG | | No CAG | | CAG | | |
| Variables, N (%) | CAD (n = 413 Pts) | No CAD (n = 261 Pts) | (n = 91 Pts) | P value | CAD (n = 160 Pts) | Non-CAD (n = 160 Pts) | P value |
| **Five-year clinical outcomes** | | | | | | | |
| Total death | 51 (12.3) | 25 (9.6) | 12 (13.2) | 0.241 | 13 (8.1) | 21 (13.1) | 0.153 |
| Cardiac death | 12 (3.0) | 2 (0.8) | 3 (3.6) | 0.066 | 5 (3.1) | 2 (1.3) | 0.263 |
| Non-Cardiac death | 39 (9.6) | 23 (8.9) | 9 (10.0) | 0.685 | 8 (5.1) | 19 (11.9) | 0.030 |
| Myocardial infarction | 12 (3.1) | 3 (1.2) | 5 (6.0) | 0.022 | 2 (1.2) | 2 (0.6) | 0.989 |
| STEMI | 5 (1.3) | 1 (0.4) | 3 (3.6) | 0.035 | 1 (0.6) | 1 (0.6) | 0.998 |
| Coronary revascularization | 36 (9.4) | 2 (0.8) | 4 (4.9) | 0.005 | 13 (8.5) | 0 (0.0) | < 0.001 |
| Strokes | 17 (4.4) | 15 (6.0) | 2 (2.3) | 0.150 | 8 (5.2) | 10 (6.5) | 0.587 |
| MACCE | 88 (21.3) | 39 (14.9) | 17 (18.7) | 0.153 | 28 (17.5) | 29 (18.1) | 0.874 |
| **Variables, N (%)** | **CAD** (n = 545 Limb) | **Non-CAD** (n = 313 Limb) | **Not CAG** (n = 110 Limb) | **P value** | **CAD** (n = 197 Limb) | **Non-CAD** (n = 201 Limb) | **P value** |
| **Peripheral revascularization** | | | | | | | |
| Target lesion revascularization | 80 (15.6) | 57 (19.3) | 21 (20.9) | 0.781 | 33 (17.3) | 37 (19.7) | 0.593 |
| Target extremity revascularization | 86 (16.8) | 62 (21.0) | 22 (21.9) | 0.627 | 36 (18.9) | 41 (21.7) | 0.491 |
| **Target extremity surgery** | | | | | | | |
| Above the knee amputations | 1 (0.1) | 3 (1.0) | 3 (2.8) | 0.366 | 0 (0.0) | 2 (1.0) | 0.157 |
| Above the ankle amputations | 23 (4.4) | 15 (4.9) | 9 (8.4) | 0.332 | 5 (2.6) | 10 (5.1) | 0.187 |
| Below the ankle amputations | 102 (19.2) | 47 (15.4) | 26 (24.4) | 0.029 | 44 (22.5) | 32 (16.4) | 0.121 |
| **Major adverse limb events** | 161 (30.6) | 93 (30.6) | 40 (38.2) | 0.226 | 65 (33.3) | 62 (32) | 0.705 |

Data are presented as incidence (%). CAD, coronary artery disease; STEMI; ST-segment elevation myocardial infarction; MACCE, major adverse cerebrovascular and cardiac events.

coronary revascularization. The non-cardiac death was higher in the non-CAD group (5.1% vs. 12.0%, P<0.001) but the coronary revascularization was higher in the CAD group (8.5% vs. 0.0%, P<0.001) (Table 4).

## Discussion

The main finding of this study is that routine CAG and subsequent PCI for significant CAD in symptomatic PAD patients undergoing PTA is safe and resulted in similar long-term survival as compared with the symptomatic PAD patients who undergoing PTA and who did not have CAD. Not surprisingly, repeat PCI was performed more frequently in PAD patients with CAD at long-term follow-up.

The summarize of this study is that PAD patients with CAD were expected to have very poor long-term survival, but they are shown no different long-term prognosis such as mortality compared to PAD patients without CAD. These PAD patients with CAD had received PCI and/or optimal medication treatment after the CAG. Considering the high mortality rate of patients with PAD [15], the imaging evaluation of the coronary arteries such as the CAG in PAD patients is important for improving their long-term survival.

PTA is commonly performed to treat PAD patients such as critical limb ischemia (CLI). Both PAD and CAD are known to be associated with a poor quality of life and a poor prognosis [1,2,5,7]. These diseases share most risk factors, and are often comorbid. The best treatment

Kaplan-Meier Analysis

Cox-proportional hazards regression analysis

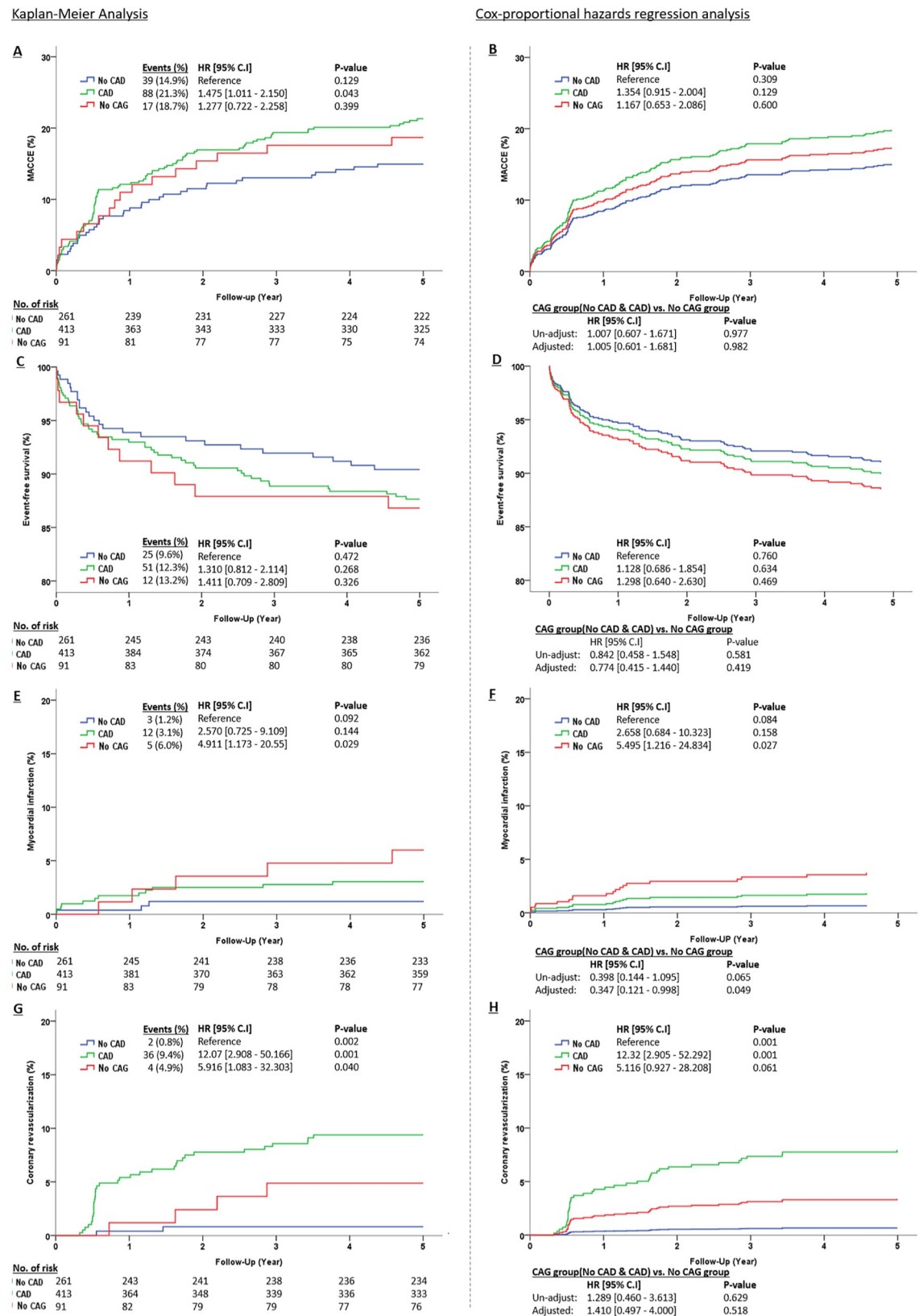

**Fig 1. Survival analysis of 5-year clinical outcomes by Kaplan–Meier curve and Cox-proportional hazards regression analysis.**
Figure A,C,E, and G show an Un-adjusted hazard ratio by Cox-proportional hazards regression analysis. Figure B,D,F, and H show an adjusted hazard ratio by Cox-proportional hazards regression analysis. Adjusted confounders are Wound, Sex, Age, Hypertension, Diabetes mellitus, Chronic Renal Insufficiency, Current of smoking, treated limb side, and treated lesion (femoral, Popliteal, and Below the knee artery). MACCE, major adverse cerebrovascular and cardiac events; CI, confidence interval; CAD, coronary artery disease; CAG, coronary angiogram.

option for these patients may be lifestyle modification and early prevention through global risk reduction [4,14,16,17]. In addition, evaluation of coronary arteries along with subsequent optimal treatment of PAD patients undergoing PTA may improve survival [6,8,11,13,18]. However, an effects on a routine evaluation of the coronary artery such as routine CAG and echocardiography has been controversial in patients undergoing PTA in terms of cost effectiveness. The majority of CAD patients who received PCI in our study received elective PCI. The COURAGE Trial shown that PCI may not have the advantage in stable CAD patients [23]. Also, some clinician can think of it, regardless of symptoms, doing PCI based on CAG results can be an excessive treatment. However, a large number of patients receiving PTA are CLI patients such as wounds and diabetic foot ulcer. Two-thirds of our study population is CLI. Most CLI patients may have limited activity, making it difficult to clearly assess coronary function and associated with less ischemic symptoms. Higher prevalence of advanced PAD patients also associated with higher incidence of silent myocardial ischemia. Our research results provide insight into the long-term clinical effects of active PCI for CAD patients according to routine evaluation of coronary artery and treatment decisions by clinician in PAD patients undergoing PTA. In real world clinical practice, physicians who are performing endovascular intervention commonly only focusing on the extremity target lesion intervention without concerning of CAD evaluation and management [15]. Also, the use of cardio-protective drugs such as antiplatelet therapy, statin, and ACE inhibitors in PAD helps to improve survival, but the introduction of these guidelines is more than a decade behind CAD [15]. Thus, main intention of this report is to provoke all the endovascular intervention specialties should check patient's co-existing significant CAD and to have an idea to safely manage the CAD together to prevent future cardiovascular events. Routine CAD checkup is not commonly widely accepted in daily clinical practice, especially in terms of cost-effectiveness but this should be changed according to our novel data.

Generally, severe CAD is reported in 54% to 69% of patients with CLI [7–10]. In the present study, 61.2% of patients with PAD were diagnosed with severe CAD. Left main disease and multi-vessel disease were observed at frequencies of 15% and 59%, respectively, in the CAD patients, which portend poor prognosis if left undiagnosed and untreated. Therefore, prompt diagnosis of life-threatening CAD and appropriate revascularization may improve clinical outcomes. Similary, Faglia et al., reported that attention to CAD at the time of admission for treatment of CLI improves the survival of patients with diabetes [11].

Aforementioned in introduction part, patients with PAD have particularly a high mortality rate from cardiovascular events [6,15]. Mortality rates of up to 20% within 6 months from diagnosis and in excess of 50% at 5 years have been reported for CLI [15]. On the other hand, in our study, the mortality rates of PTA patients were significantly lower than the previous studies. The present observation of long-term clinical results of 5-years in patients undergoing PTA, mortality rates were similar among the CAD group, the non-CAD group and the no-CAG group, respectively (12.3% vs. 9.6 vs. 13.2%). This means that the imaging evaluation of the coronary arteries in PAD patients is important for improving their long-term survival and prevent of MI.

After PSM analysis, coronary revascularization remained higher in the CAD group than the non-CAD group, highlighting the fact that repeat coronary revascularization remains a problem in CAD patients. Also, regardless of the baseline risk adjustment, there was no difference between the two groups in PTA-related events such as peripheral revascularization or target extremity surgery. The strategy for CAD evaluation and treatment in PTA patients seems to be a safe and effective strategy not only for better short-term outcomes but also durable long-term outcomes. Similar to our research, Chen and colleagues have registered the multi-center randomized controlled trial (NCT02169258) "Routine Coronary Catheterization in Low Extremity Artery Disease Undergoing Percutaneous Transluminal Angioplasty (PIROUET-TE-PTA)" at ClinicalTrials.gov [18]. Estimated enrollment is 700 participants and the primary endpoint of the study was composite major adverse cardiac events at the 1-year follow-up.

There are some limitations to our study. First, the results of our study were derived indirectly by comparison with the survival rates of other studies. In general, the 5-year survival rate of PAD patients is around 50% [15], and the survival rate of our study subjects, 88.5%, was very high. The no CAG group without cardiac function evaluation should be allocated to verify the effectiveness of routine CAG in PAD patients, but this is a problem that may be against research ethics. In this study, PTA was performed by an interventionist based on cardiology, the cardiac function of all subjects was evaluated non-invasively and/or invasively by the cardiologist. Therefore, the no CAG group in this study should not be confused with the patients that did not perform cardiac function evaluation. Second, we analyzed data retrospectively, and PSM analysis was performed to minimize confounding factors, which could have affected our results. The registry was designed as an all-comers prospective registry from 2006. However, we could not adjust for all limiting factors not shown in medical records or collected through telephone contact. Third, in the PSM analysis, a total of 320 patients from 160 pairs in both groups were analyzed. This sample size may be insufficient to produce results. Our study is a registered observational study and may be a limitation of analysis. As Chen and colleagues' research results come out, the results will be updated a bit (18).

In conclusion, a strategy of routine CAG and subsequent PCI, if required, appears to be a reasonable strategy for significant PAD patients undergoing PTA. Our results highlight the importance of CAD evaluation in patients with PAD. A result of randomized trial is needed to assess the efficacy and safety of this treatment strategy for PAD patients finally [18].

## Author Contributions

**Conceptualization:** Ji-Yeon Hong, Seung-Woon Rha.

**Data curation:** Byoung Geol Choi, Seung-Woon Rha.

**Formal analysis:** Byoung Geol Choi, Ji-Yeon Hong, Cheol Ung Choi.

**Investigation:** Seung-Woon Rha, Cheol Ung Choi, Michael S. Lee.

**Methodology:** Byoung Geol Choi, Ji-Yeon Hong.

**Project administration:** Seung-Woon Rha.

**Resources:** Byoung Geol Choi, Seung-Woon Rha.

**Software:** Byoung Geol Choi.

**Supervision:** Michael S. Lee.

**Validation:** Byoung Geol Choi, Ji-Yeon Hong, Michael S. Lee.

**Writing – original draft:** Byoung Geol Choi, Ji-Yeon Hong.

**Writing – review & editing:** Byoung Geol Choi, Ji-Yeon Hong, Seung-Woon Rha, Cheol Ung Choi, Michael S. Lee.

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
