## [Decision Letter · Decision Letter 0]

15 Oct 2019

PONE-D-19-21312

Long-term Outcomes of Peripheral Arterial Disease Patients with Significant Coronary Artery Disease underwent Percutaneous Coronary Intervention

PLOS ONE

Dear Prof. Rha

Thank you for submitting your manuscript to PLOS ONE. After careful consideration, we feel that it has merit but does not fully meet PLOS ONE’s publication criteria as it currently stands. Therefore, we invite you to submit a revised version of the manuscript that addresses the points raised during the review process.

We would appreciate receiving your revised manuscript by December 10, 2019. To enhance the reproducibility of your results, we recommend that if applicable you deposit your laboratory protocols in protocols.io, where a protocol can be assigned its own identifier (DOI) such that it can be cited independently in the future. For instructions see: http://journals.plos.org/plosone/s/submission-guidelines#loc-laboratory-protocols

We look forward to receiving your revised manuscript.

Kind regards,

Xianwu Cheng, M.D., Ph.D., FAHA

Academic Editor

PLOS ONE

Journal Requirements:

2) During your revisions, please note that a simple title correction is required: "Long-term Outcomes of Peripheral Arterial Disease Patients with Significant Coronary Artery Disease *undergoing* Percutaneous Coronary Intervention". Please ensure this is updated in the manuscript file and the online submission information.

3) Thank you for stating the following financial disclosure:

 [The authors have no financial conflicts of interest relevant to the manuscript to

disclose.].

Please provide an amended Funding Statement that declares *all* the funding or sources of support received during this specific study (whether external or internal to your organization) as detailed online in our guide for authors at http://journals.plos.org/plosone/s/submit-now.  Please state what role the funders took in the study.  If any authors received a salary from any of your funders, please state which authors and which funder. If the funders had no role, please state: "The funders had no role in study design, data collection and analysis, decision to publish, or preparation of the manuscript."

4)  In your Data Availability statement, you have not specified where the minimal data set underlying the results described in your manuscript can be found. PLOS defines a study's minimal data set as the underlying data used to reach the conclusions drawn in the manuscript and any additional data required to replicate the reported study findings in their entirety. All PLOS journals require that the minimal data set be made fully available. For more information about our data policy, please see http://journals.plos.org/plosone/s/data-availability.

Reviewers' comments:

Reviewer's Responses to Questions

**Comments to the Author**

1. Is the manuscript technically sound, and do the data support the conclusions?

Reviewer #1: Yes

Reviewer #2: Yes

2. Has the statistical analysis been performed appropriately and rigorously? 

Reviewer #1: No

Reviewer #2: Yes

3. Have the authors made all data underlying the findings in their manuscript fully available?

Reviewer #1: Yes

Reviewer #2: Yes

4. Is the manuscript presented in an intelligible fashion and written in standard English?

Reviewer #1: Yes

Reviewer #2: Yes

5. Review Comments to the Author

Reviewer #1: The main finding of this study is that routine coronary angiography and subsequent percutaneous intervention for significant coronary artery disease in patients with symptomatic peripheral arterial disease is safe and resulted in similar long-term survival to those who did not have coronary artery disease.

1 The correlation of coronary and peripheral arterial disease is well known and treatment strategies have been discussed and examined. What therefore is the novelty in this study of its conclusions

2 How does a single center study that performed angiography on each patients help

a. Would it not have been more meaningful to compare those with symptoms who did and did not undergo angiography

3 What new conclusions were learned not seen in ref, 8 the authors 1 year experience with this group

4 Was a power calculation performed to determine if 160 matched patients could indeed detect a meaningful difference

Reviewer #2: Comments to Authors:

The authors provide an interesting and potential important manuscript describing"Long-term Outcomes of Peripheral Arterial Disease Patients with Significant Coronary Artery Disease underwent Percutaneous Coronary Intervention" The main issues concerning this paper are those concerning the relationship between PAD and CAD. The previous study had figured out that 54%-69% of CAD patients with PAD, but a limited paper had researched that the percent of CAD in PAD patients.This study was designed to observe CAD in patients with PAD, and conclude that patients with PAD often have CAD disease.

There are some weak points that need to be addressed by the authors.

Major

1. In all patient analysis, the CAD patients showed a significantly higher prevalence of DM as compared with non-CAD subjects (P < 0.01). However, in age-matched analysis, there was no significant difference between two groups. The authors explain it.

2. As shown in Table 4, in all patients, total death (five-year outcomes) was higher in CAD group than in non-CAD group without significant difference. In contrast, in age matched analysis, it was higher non-CAD group than that of CDA patients. Why?

3. Discussion sections were disorganized, the authors did not clearly figured out their own findings and did not derive clinical implication in each section based on your own and previous observations..

6. PLOS authors have the option to publish the peer review history of their article (what does this mean?). If published, this will include your full peer review and any attached files.

Reviewer #1: No

Reviewer #2: Yes: Hailong Wang

---

## [Author Response · Author response to Decision Letter 0]

23 Mar 2020

PONE-D-19-21312

Long-term Outcomes of Peripheral Arterial Disease Patients with Significant Coronary Artery Disease undergoing Percutaneous Coronary Intervention

Dear editor: 

 Thank you so much for your e-mail sent to us on November 29 2019. We really appreciate the great opportunity you have kindly given to us to resubmit our work!! We really appreciate your excellent advice which has helped us a lot to improve the quality of our manuscript. According to your comments, we have amended the relevant parts in the manuscript. All of your questions were answered below. In the process of revising the manuscript, Jin-Seok Kim was added as an author, and he was fully involved in the revision of the study. All authors have approved Dr. Kim's inclusion as an author.

Yours sincerely,

*Correspondence to: 

Seung-Woon Rha, MD, PhD, E‐mail: swrha617@yahoo.co.kr (S-WR)

Byoung Geol Choi, PhD, E‐mail: trv940@korea.ac.kr (BGC)

Journal Requirements:

Response: Thank you so much for your careful review. We reflected this in the revised manuscript.

2) During your revisions, please note that a simple title correction is required: "Long-term Outcomes of Peripheral Arterial Disease Patients with Significant Coronary Artery Disease *undergoing* Percutaneous Coronary Intervention". Please ensure this is updated in the manuscript file and the online submission information.

Response: Thank you so much for your careful review and comment. We reflected this in the title of the revised manuscript as following as “Long-term Outcomes of Peripheral Arterial Disease Patients with Significant Coronary Artery Disease undergoing Percutaneous Coronary Intervention”

3) Thank you for stating the following financial disclosure:

 [The authors have no financial conflicts of interest relevant to the manuscript to

disclose.].

Please provide an amended Funding Statement that declares *all* the funding or sources of support received during this specific study (whether external or internal to your organization) as detailed online in our guide for authors at http://journals.plos.org/plosone/s/submit-now. 

Please state what role the funders took in the study. If any authors received a salary from any of your funders, please state which authors and which funder. If the funders had no role, please state: "The funders had no role in study design, data collection and analysis, decision to publish, or preparation of the manuscript."

Response: Thank you so much for your careful review and comment. We reflected this in the revised manuscript (line 18 on page 1 to line 22) and the cover letter.

4) In your Data Availability statement, you have not specified where the minimal data set underlying the results described in your manuscript can be found. PLOS defines a study's minimal data set as the underlying data used to reach the conclusions drawn in the manuscript and any additional data required to replicate the reported study findings in their entirety. All PLOS journals require that the minimal data set be made fully available. For more information about our data policy, please see http://journals.plos.org/plosone/s/data-availability.

Response: Thank you so much for your careful review and comment. We reflected this in the revised manuscript (line 21 on page 1).

Dear reviewer:

Thank you so much for your very insightful and serious comments for our manuscript! We really appreciate your excellent advice which has helped us a lot in improving our manuscript. According to your comments, we have amended the relevant parts in the manuscript. Some of your questions were answered below.

Yours sincerely,

*Correspondence to: 

Seung-Woon Rha, MD, PhD, E‐mail: swrha617@yahoo.co.kr (S-WR)

Byoung Geol Choi, PhD, E‐mail: trv940@korea.ac.kr (BGC)

Reviewer #1: The main finding of this study is that routine coronary angiography and subsequent percutaneous intervention for significant coronary artery disease in patients with symptomatic peripheral arterial disease is safe and resulted in similar long-term survival to those who did not have coronary artery disease.

1 The correlation of coronary and peripheral arterial disease is well known and treatment strategies have been discussed and examined. What therefore is the novelty in this study of its conclusions

Response: Thank you so much for your careful review and important comment. As you understand, the treatment of PAD and CAD is very important for improving patient’s survival and quality of life. However, in patients undergoing PTA, routine evaluation of the presence of significant coronary artery disease may be controversial. In real world clinical practice, physicians who are performing endovascular intervention commonly only focusing on the extremity target lesion intervention without concerning of coronary artery disease evaluation and management, especially in particular specialties such as vascular surgeon or interventional radiologists. Thus, main intention of this report is to provoke all the endovascular intervention specialties should check patient’s co-existing significant CAD and to have an idea to safely manage the CAD together to prevent future cardiovascular events. Routine CAD checkup is not commonly widely accepted in daily clinical practice, especially in terms of cost-effectiveness but this should be changed according to our novel data.

Therefore, we added the following in the second paragraph with a conclusion in the last paragraph of the discussion part (line 1 on page 10 to line 4)

2 How does a single center study that performed angiography on each patients help

a. Would it not have been more meaningful to compare those with symptoms who did and did not undergo angiography

Response: Thank you so much for your careful review and important comment. Korea University Guro Hospital (KUGH), which conducted this study, is a university hospital for training, and is a large-scale center that performs many CAG, PCI, and PTAs every year. Data are collected by a trained research coordinator using a standardized case report form. And professional researchers analyze it consistently. It may be smaller than a multicenter study, but it can collect and analyze data more efficiently and accurately. Our study is a comparative study of the CAD group and the non-CAD group who received appropriate treatment through coronary artery evaluation in patients undergoing PTA. Therefore, the group without CAG was excluded. Main purpose of this study is to show whether the concomitant optimal revascularization of significant CAD regardless of patient’s symptom in PAD patients undergoing PTA could reduce the cardiovascular event risk compared with PAD patients without significant CAD. Selective CAD evaluation according to ischemic symptoms in PAD patients undergoing PTA is not a suitable and safe strategy because many of them are elderly, diabetics and limited ambulation to provoke ischemic chest pain or dyspnea due to diabetic foot wound. Thus, selective symptom driven CAD evaluation cannot be ideally accepted in context of ‘pan-atherosclerosis’ of PAD patients. We will consider a comparative study in symptomatic PAD patients between CAG and non-CAG groups in future studies.

3 What new conclusions were learned not seen in ref, 8 the authors 1-year experience with this group

Response: Thank you so much for your careful review and important comment. The results of our previous study, which observed the results for one year, the CAD group that received optimal treatment through routine coronary artery evaluation showed similar clinical results than the non-CAD group in CLI patients received PTA [8]. Similarly, in the present observation of long-term clinical results of 5-year in patients undergoing PTA, the clinical results of the CAD and non-CAD groups were similar. The strategy for CAD evaluation and treatment in PTA patients seems to be a safe and effective strategy not only for better short-term outcomes but also durable long-term outcomes. We wanted to show the durable long-term results from this routine CAD work up and management strategy in PAD patients undergoing PTA. We added this to the discussion part (line 20 on page 9 to line 14 on page 10). 

4 Was a power calculation performed to determine if 160 matched patients could indeed detect a meaningful difference

Response: Thank you so much for your careful review and important comment. This study was designed as a prospective registration study and retrospective analysis was performed. Therefore, the sample size after PSM may not be sufficient, which may be a limitation of the study. We added this to the limitations of the discussion part (line 17 on page 10 to line 19) as following as “Second, In the PSM analysis, a total of 320 patients from 160 pairs in both groups were analyzed. This sample size may be insufficient to produce results. Our study is a registered observational study and may be a limitation of analysis.”

Dear reviewer:

Thank you so much for your very insightful and serious comments for our manuscript! We really appreciate your excellent advice which has helped us a lot in improving our manuscript. According to your comments, we have amended the relevant parts in the manuscript. Some of your questions were answered below.

Yours sincerely,

*Correspondence to: 

Seung-Woon Rha, MD, PhD, E‐mail: swrha617@yahoo.co.kr (S-WR)

Byoung Geol Choi, PhD, E‐mail: trv940@korea.ac.kr (BGC)

Reviewer #2: Comments to Authors:

The authors provide an interesting and potential important manuscript describing "Long-term Outcomes of Peripheral Arterial Disease Patients with Significant Coronary Artery Disease underwent Percutaneous Coronary Intervention" The main issues concerning this paper are those concerning the relationship between PAD and CAD. The previous study had figured out that 54%-69% of CAD patients with PAD, but a limited paper had researched that the percent of CAD in PAD patients. This study was designed to observe CAD in patients with PAD, and conclude that patients with PAD often have CAD disease.

There are some weak points that need to be addressed by the authors.

Major

1. In all patient analysis, the CAD patients showed a significantly higher prevalence of DM as compared with non-CAD subjects (P < 0.01). However, in age-matched analysis, there was no significant difference between two groups. The authors explain it.

Response: Thank you so much for your careful review and important comment. In this study, the CAD group had a higher prevalence of DM than the non-CAD. DM, like CAD, is an important and independent factor affecting the survival of PAD patients. The propensity score matching used in this study is a method of adjusting all the possible baseline confounding variables such as DM, hypertension, smoking, alcohol drinking, laboratory findings, and others that can affect not only the age but also the endpoints of this study. 

2. As shown in Table 4, in all patients, total death (five-year outcomes) was higher in CAD group than in non-CAD group without significant difference. In contrast, in age matched analysis, it was higher non-CAD group than that of CAD patients. Why?

Response: Thank you so much for your careful review and important comment. In all patient analysis of this study, the CAD group had more elderly, diabetic, and higher creatinine levels than patients in the non-CAD group (Table 1). Although not statistically significant, this baseline difference can explain the higher mortality rate in the CAD group. After PSM analysis (This is not an age matching method), although there was no statistical significance, in contrast, the non-CAD group had a high mortality rate. This is because non-cardiac death was higher in the non-CAD group than in the CAD group. As in "Matched patients" in Tables 1 to 3, the baseline characteristics of the two groups were well balanced by the PSM method, but the variables that could not be evaluated may have affected non-cardiac death. This is described in the limitations of the discussion part (line 7 on page 10 to line 9) as follows as “we could not adjust for all limiting factors not shown in medical records or collected through telephone contact.”

3. Discussion sections were disorganized, the authors did not clearly figured out their own findings and did not derive clinical implication in each section based on your own and previous observations.

Response: Thank you so much for your careful review and important comment. We actively revised the discussion to reflect your opinion. We attach a revised manuscript.

---

## [Decision Letter · Decision Letter 1]

13 May 2020

PONE-D-19-21312R1

Long-term Outcomes of Peripheral Arterial Disease Patients with Significant Coronary Artery Disease undergoing Percutaneous Coronary Intervention

PLOS ONE

Dear Prof. Rha

Thank you for submitting your manuscript to PLOS ONE. After careful consideration, we feel that it has merit but does not fully meet PLOS ONE’s publication criteria as it currently stands. Therefore, we invite you to submit a revised version of the manuscript that addresses the points raised during the review process.

We would appreciate receiving your revised manuscript by Jun 27 2020 11:59PM. To enhance the reproducibility of your results, we recommend that if applicable you deposit your laboratory protocols in protocols.io, where a protocol can be assigned its own identifier (DOI) such that it can be cited independently in the future. For instructions see: http://journals.plos.org/plosone/s/submission-guidelines#loc-laboratory-protocols

We look forward to receiving your revised manuscript.

Kind regards,

Xianwu Cheng, M.D., Ph.D., FAHA

Academic Editor

PLOS ONE

Additional Editor Comments (if provided):

Reviewer#1 has still pointed out the original concerns was not addressed satisfactory by the authors. Thus, the authors may resubmit a revised version one more time, but it will be re-reviewed and there exists no guarantee that even with revision it will necessarily be accepted.

Reviewers' comments:

Reviewer's Responses to Questions

**Comments to the Author**

1. If the authors have adequately addressed your comments raised in a previous round of review and you feel that this manuscript is now acceptable for publication, you may indicate that here to bypass the “Comments to the Author” section, enter your conflict of interest statement in the “Confidential to Editor” section, and submit your "Accept" recommendation.

Reviewer #1: (No Response)

Reviewer #2: All comments have been addressed

2. Is the manuscript technically sound, and do the data support the conclusions?

Reviewer #1: No

Reviewer #2: Yes

3. Has the statistical analysis been performed appropriately and rigorously? 

Reviewer #1: No

Reviewer #2: Yes

4. Have the authors made all data underlying the findings in their manuscript fully available?

Reviewer #1: Yes

Reviewer #2: Yes

5. Is the manuscript presented in an intelligible fashion and written in standard English?

Reviewer #1: Yes

Reviewer #2: Yes

6. Review Comments to the Author

Reviewer #1: I will restrict my comments to responses on my previous critique

Reviewer #1: The main finding of this study is that routine coronary angiography and subsequent percutaneous intervention for significant coronary artery disease in patients with symptomatic peripheral arterial disease is safe and resulted in similar long-term survival to those who did not have coronary artery disease.

1 The correlation of coronary and peripheral arterial disease is well known and treatment strategies have been discussed and examined. What therefore is the novelty in this study of its conclusions

Response: Thank you so much for your careful review and important comment. As you understand, the treatment of PAD and CAD is very important for improving patient’s survival and quality of life. However, in patients undergoing PTA, routine evaluation of the presence of significant coronary artery disease may be controversial. In real world clinical practice, physicians who are performing endovascular intervention commonly only focusing on the extremity target lesion intervention without concerning of coronary artery disease evaluation and management, especially in particular specialties such as vascular surgeon or interventional radiologists.

ARE THERE ANY DATA TO SUPPORT THIS STATEMENT. WE ARE IN FACT CONCERNED ABOUT THE PRESENCE OF CAD IN PAD PATIENTS IN OUR INSTITUTION AND ELSEWHERE.

Thus, main intention of this report is to provoke all the endovascular intervention specialties should check patient’s co-existing significant CAD and to have an idea to safely manage the CAD together to prevent future cardiovascular events. Routine CAD checkup is not commonly widely accepted in daily clinical practice, especially in terms of cost-effectiveness but this should be changed according to our novel data.

Therefore, we added the following in the second paragraph with a conclusion in the last paragraph of the discussion part (line 1 on page 10 to line 4)

2 How does a single center study that performed angiography on each patients help

a. Would it not have been more meaningful to compare those with symptoms who did and did not undergo angiography

Response: Thank you so much for your careful review and important comment. Korea University Guro Hospital (KUGH), which conducted this study, is a university hospital for training, and is a large-scale center that performs many CAG, PCI, and PTAs every year. Data are collected by a trained research coordinator using a standardized case report form. And professional researchers analyze it consistently. It may be smaller than a multicenter study, but it can collect and analyze data more efficiently and accurately. Our study is a comparative study of the CAD group and the non-CAD group who received appropriate treatment through coronary artery evaluation in patients undergoing PTA. Therefore, the group without CAG was excluded. Main purpose of this study is to show whether the concomitant optimal revascularization of significant CAD regardless of patient’s symptom in PAD patients undergoing PTA could reduce the cardiovascular event risk compared with PAD patients without significant CAD. Selective CAD evaluation according to ischemic symptoms in PAD patients undergoing PTA is not a suitable and safe strategy because many of them are elderly, diabetics and limited ambulation to provoke ischemic chest pain or dyspnea due to diabetic foot wound. Thus, selective symptom driven CAD evaluation cannot be ideally accepted in context of ‘pan-atherosclerosis’ of PAD patients. We will consider a comparative study in symptomatic PAD patients between CAG and non-CAG groups in future studies.

THE FAILURE TO COMPARE THOSE WHO DID WITH THOSE WHO DID NOT UNDERGO ANGIOGRAPHY LEAVES NO DEFINITIVE CONCLUSION

3 What new conclusions were learned not seen in ref, 8 the authors 1-year experience with this group

Response: Thank you so much for your careful review and important comment. The results of our previous study, which observed the results for one year, the CAD group that received optimal treatment through routine coronary artery evaluation showed similar clinical results than the non-CAD group in CLI patients received PTA [8]. Similarly, in the present observation of long-term clinical results of 5-year in patients undergoing PTA, the clinical results of the CAD and non-CAD groups were similar. The strategy for CAD evaluation and treatment in PTA patients seems to be a safe and effective strategy not only for better short-term outcomes but also durable long-term outcomes. We wanted to show the durable long-term results from this routine CAD work up and management strategy in PAD patients undergoing PTA. We added this to the discussion part (line 20 on page 9 to line 14 on page 10).

WHY WOULD ONE EXPECT THAT LONG-TERM OUTCOMES WOULD BE SIGNIFICANTLY WORSE THAN SHORT TERM. THIS IS WHY THERE NEEDS TO BE A NON-ANGIOGRAM GROUP TO SEE NOT ONLY OUTCOME BUT CROSSOVER

4 Was a power calculation performed to determine if 160 matched patients could indeed detect a meaningful difference

Response: Thank you so much for your careful review and important comment. This study was designed as a prospective registration study and retrospective analysis was performed. Therefore, the sample size after PSM may not be sufficient, which may be a limitation of the study. We added this to the limitations of the discussion part (line 17 on page 10 to line 19) as following as “Second, In the PSM analysis, a total of 320 patients from 160 pairs in both groups were analyzed. This sample size may be insufficient to produce results. Our study is a registered observational study and may be a limitation of analysis.”

PERHAPS THE AUTHORS COULD ADD A STATEMENT AS TO HOW MANY PATIENTS WOULD HAVE NEEDED TO BE STUDIED TO SHOW A DIFFERENCE RATHER THAN SAY MAY BE INSUFFICIENT.

Reviewer #2: Thank you very much for your reasonable explanation. I believe this paper will give readers a new perspective on CVD and PAD

7. PLOS authors have the option to publish the peer review history of their article (what does this mean?). If published, this will include your full peer review and any attached files.

Reviewer #1: No

Reviewer #2: No

---

## [Author Response · Author response to Decision Letter 1]

25 Oct 2020

Dear reviewer:

Thank you so much for your very insightful and serious comments for our manuscript! We really appreciate your excellent advice which has helped us a lot in improving our manuscript. According to your comments, we have amended the relevant parts in the manuscript. Some of your questions were answered below.

Yours sincerely,

*Correspondence to: 

Seung-Woon Rha, MD, PhD, E‐mail: swrha617@yahoo.co.kr

Reviewer #1: The main finding of this study is that routine coronary angiography and subsequent percutaneous intervention for significant coronary artery disease in patients with symptomatic peripheral arterial disease is safe and resulted in similar long-term survival to those who did not have coronary artery disease.

1 The correlation of coronary and peripheral arterial disease is well known and treatment strategies have been discussed and examined. What therefore is the novelty in this study of its conclusions

Response: Thank you so much for your careful review and important comment. As you understand, the treatment of PAD and CAD is very important for improving patient’s survival and quality of life. However, in patients undergoing PTA, routine evaluation of the presence of significant coronary artery disease may be controversial. In real world clinical practice, physicians who are performing endovascular intervention commonly only focusing on the extremity target lesion intervention without concerning of coronary artery disease evaluation and management, especially in particular specialties such as vascular surgeon or interventional radiologists.

ARE THERE ANY DATA TO SUPPORT THIS STATEMENT. WE ARE IN FACT CONCERNED ABOUT THE PRESENCE OF CAD IN PAD PATIENTS IN OUR INSTITUTION AND ELSEWHERE.

Response: Thank you so much for your careful review and important comment. As you know that the patients with PAD have particularly a high mortality rate from cardiovascular events. The severe CAD has been reported in 54% to 69% of patients with PAD [1-4]. So, the use of cardio-protective drugs as antiplatelet therapy, statin, and ACE inhibitors in PAD helps improve survival, but the introduction of these guidelines is more than a decade behind CAD. Also, the underuse of cardio-protective medication in the PAD population in comparison to patients with CAD has been published previously. These are well explained in Teraa's review. We cited his review in the revised manuscript [5].

Therefore, we changed the following in the second paragraph of the discussion part (line 4 to line 8 on page 11) as follows as “ In real world clinical practice, physicians who are performing endovascular intervention commonly only focusing on the extremity target lesion intervention without concerning of CAD evaluation and management. Also, the use of cardio-protective drugs such as antiplatelet therapy, statin, and ACE inhibitors in PAD helps to improve survival, but the introduction of these guidelines is more than a decade behind CAD.”

[References]

1. Faglia E, Dalla Paola L, Clerici G, Clerissi J, Graziani L, et al. (2005) Peripheral angioplasty as the first-choice revascularization procedure in diabetic patients with critical limb ischemia: prospective study of 993 consecutive patients hospitalized and followed between 1999 and 2003. Eur J Vasc Endovasc Surg 29: 620-627.

2. Lee MS, Rha SW, Han SK, Choi BG, Choi SY, et al. (2015) Clinical outcomes of patients with critical limb ischemia who undergo routine coronary angiography and subsequent percutaneous coronary intervention. J Invasive Cardiol 27: 213-217.

3. Nishijima A, Yamamoto N, Yoshida R, Hozawa K, Yanagibayashi S, et al. (2017) Coronary Artery Disease in Patients with Critical Limb Ischemia Undergoing Major Amputation or Not. Plast Reconstr Surg Glob Open 5: e1377.

4. Ryu HM, Kim JS, Ko YG, Hong MK, Jang Y, et al. (2012) Clinical outcomes of infrapopliteal angioplasty in patients with critical limb ischemia. Korean Circ J 42: 259-265.

5. Teraa M, Conte MS, Moll FL, Verhaar MC (2016) Critical Limb Ischemia: Current Trends and Future Directions. J Am Heart Assoc 5.

2 How does a single center study that performed angiography on each patients help

a. Would it not have been more meaningful to compare those with symptoms who did and did not undergo angiography

Response: Thank you so much for your careful review and important comment. Korea University Guro Hospital (KUGH), which conducted this study, is a university hospital for training, and is a large-scale center that performs many CAG, PCI, and PTAs every year. Data are collected by a trained research coordinator using a standardized case report form. And professional researchers analyze it consistently. It may be smaller than a multicenter study, but it can collect and analyze data more efficiently and accurately. Our study is a comparative study of the CAD group and the non-CAD group who received appropriate treatment through coronary artery evaluation in patients undergoing PTA. Therefore, the group without CAG was excluded. Main purpose of this study is to show whether the concomitant optimal revascularization of significant CAD regardless of patient’s symptom in PAD patients undergoing PTA could reduce the cardiovascular event risk compared with PAD patients without significant CAD. Selective CAD evaluation according to ischemic symptoms in PAD patients undergoing PTA is not a suitable and safe strategy because many of them are elderly, diabetics and limited ambulation to provoke ischemic chest pain or dyspnea due to diabetic foot wound. Thus, selective symptom driven CAD evaluation cannot be ideally accepted in context of ‘pan-atherosclerosis’ of PAD patients. We will consider a comparative study in symptomatic PAD patients between CAG and non-CAG groups in future studies.

THE FAILURE TO COMPARE THOSE WHO DID WITH THOSE WHO DID NOT UNDERGO ANGIOGRAPHY LEAVES NO DEFINITIVE CONCLUSION

Response: Thank you so much for your important comment. We fully agree with your opinion. In the revised manuscript, we analyzed by adding 91 cases (as defined as the no-CAG group) who did not perform routine coronary angiography. Patients were divided into three groups; 1) routine CAG and a presence of CAD (CAD group: 413 patients), 2) routine CAG without CAD (non-CAD group: 261 patients), and 3) no-CAG group (91 patients). During the 5-year clinical follow-up, the routine CAG group was associated with reduced risk of myocardial infarction by 65.3% as compared to the no-CAG group. Also, there were no differences in the incidence of any clinical events between the CAD group and the non-CAD group except for the incidence of repeat percutaneous coronary intervention (PCI), which was higher in the CAD group than the non-CAD group. These are the results section of the revised manuscript (lines 19 to 19 lines on 8 pages, 15 to 20 lines on 9 pages), the discussion section (lines 22 to 24 lines on 11 pages) and the statistical analysis part (lines 19 to 21 on 7 page) in the method section. 

3 What new conclusions were learned not seen in ref, 8 the authors 1-year experience with this group

Response: Thank you so much for your careful review and important comment. The results of our previous study, which observed the results for one year, the CAD group that received optimal treatment through routine coronary artery evaluation showed similar clinical results than the non-CAD group in CLI patients received PTA [8]. Similarly, in the present observation of long-term clinical results of 5-year in patients undergoing PTA, the clinical results of the CAD and non-CAD groups were similar. The strategy for CAD evaluation and treatment in PTA patients seems to be a safe and effective strategy not only for better short-term outcomes but also durable long-term outcomes. We wanted to show the durable long-term results from this routine CAD work up and management strategy in PAD patients undergoing PTA. We added this to the discussion part (line 20 on page 9 to line 14 on page 10).

WHY WOULD ONE EXPECT THAT LONG-TERM OUTCOMES WOULD BE SIGNIFICANTLY WORSE THAN SHORT TERM. THIS IS WHY THERE NEEDS TO BE A NON-ANGIOGRAM GROUP TO SEE NOT ONLY OUTCOME BUT CROSSOVER

Response: Thank you so much for your important comment. We fully agree with your opinion. As mentioned earlier, we analyzed by adding A NON-ANGIOGRAM GROUP (the no-CAG group) and reflected it in the revised manuscript. 

4 Was a power calculation performed to determine if 160 matched patients could indeed detect a meaningful difference

Response: Thank you so much for your careful review and important comment. This study was designed as a prospective registration study and retrospective analysis was performed. Therefore, the sample size after PSM may not be sufficient, which may be a limitation of the study. We added this to the limitations of the discussion part (line 17 on page 10 to line 19) as following as “Second, In the PSM analysis, a total of 320 patients from 160 pairs in both groups were analyzed. This sample size may be insufficient to produce results. Our study is a registered observational study and may be a limitation of analysis.”

PERHAPS THE AUTHORS COULD ADD A STATEMENT AS TO HOW MANY PATIENTS WOULD HAVE NEEDED TO BE STUDIED TO SHOW A DIFFERENCE RATHER THAN SAY MAY BE INSUFFICIENT.

Response: Thank you so much for your careful review and important comment. Already, a randomized clinical trial similar to ours is in progress, and relevant information is described in the discussion part (lines 12 to 17 lines on 12 pages) as follows as “Similar to our research, Chen and colleagues have registered the multi-center randomized controlled trial (NCT02169258) “Routine Coronary Catheterization in Low Extremity Artery Disease Undergoing Percutaneous Transluminal Angioplasty (PIROUETTE-PTA)” at ClinicalTrials.gov [18]. Estimated enrollment is 700 participants and the primary endpoint of the study was composite major adverse cardiac events at the 1-year follow-up.”

Also, we change this to the limitations part of the discussion section (line 17 to line 20 on page 12) as follows as “Second, In the PSM analysis, a total of 320 patients from 160 pairs in both groups were analyzed. This sample size may be insufficient to produce results. Our study is a registered observational study and may be a limitation of analysis. As Chen and colleagues' research results come out, the results will be updated a bit.”

---

## [Decision Letter · Decision Letter 2]

4 Nov 2020

PONE-D-19-21312R2

Long-term Outcomes of Peripheral Arterial Disease Patients with Significant Coronary Artery Disease undergoing Percutaneous Coronary Intervention

PLOS ONE

Dear Dr. Rha

Thank you for submitting your manuscript to PLOS ONE. After careful consideration, we feel that it has merit but does not fully meet PLOS ONE’s publication criteria as it currently stands. Therefore, we invite you to submit a revised version of the manuscript that addresses the points raised during the review process.

We look forward to receiving your revised manuscript.

Kind regards,

Xianwu Cheng, M.D., Ph.D., FAHA

Academic Editor

PLOS ONE

Additional Editor Comments (if provided):

The original reviewer #1 has still pointed out that the authors did not satisfactory address the original comments. He/She has concerned the statistical anapysis and data. As known, it is third peer-reviewer processes. Thus, it is final chance to revise manuscript with additional analysis and respond satisfactory to all of his/her comments.

Reviewers' comments:

Reviewer's Responses to Questions

**Comments to the Author**

1. If the authors have adequately addressed your comments raised in a previous round of review and you feel that this manuscript is now acceptable for publication, you may indicate that here to bypass the “Comments to the Author” section, enter your conflict of interest statement in the “Confidential to Editor” section, and submit your "Accept" recommendation.

Reviewer #1: (No Response)

Reviewer #2: All comments have been addressed

2. Is the manuscript technically sound, and do the data support the conclusions?

Reviewer #1: No

Reviewer #2: Yes

3. Has the statistical analysis been performed appropriately and rigorously? 

Reviewer #1: No

Reviewer #2: Yes

4. Have the authors made all data underlying the findings in their manuscript fully available?

Reviewer #1: Yes

Reviewer #2: (No Response)

5. Is the manuscript presented in an intelligible fashion and written in standard English?

Reviewer #1: No

Reviewer #2: Yes

6. Review Comments to the Author

Reviewer #1: I find this paper somewhat confusing still. The authors have tried to respond to my inquiries but as best as I can tell there is a barely statistically significant difference in groups that did and did not undergo angiography in the setting of symptomatic peripheral arterial disease.

This remains in my mind a not compelling and in fact a confusing story

I this remains confusing – there are three groups – two that had routine angiography one with and one without coronary artery disease and a third who did not undergo angiography. And the five year clinical follow up is that both groups who underwent coronary angiography were equivalent but had minor HR 0.347 p=0.049 reduction compared to those who did not. In other words regardless as to whether coronary artery disease was discovered and regardless as to whether an intervention was performed.

Is this correct?

If this is correct – what is the message that angiography is to be performed

Is it the case that there is benefit on the crossover group only or the entire group?

II Moreover, the text is till confusing – take for example the next to last paragraph which reads

Third, all subjects in this study underwent CAG, and therefore our results are not generalizable to patients who do not receive CAG. Finally, all the PAD patients with significant CAD patients did not undergone PCI due to some reasons including clinical judgment based upon the discretion of the clinician and patient preference. Only 71.6% of all CAD patients underwent PCI or CABG before, after or at the same time as admission for PTA. This registry reflects the real-world practice of PAD patients.

1 the text is unclear – there are some extra words

2 was the benefit referred to in all patients who had CAG irrespective of whether they

a. had CAD

b. had an intervention

or were only the 71.6% of all CAD patients who underwent PCI or CABG considered?

This latter issue is further confusing by the note on page “In the present study, the CAG group had reduced significantly the risk of myocardial infarction by 65.3% than the no-CAG group during 5 years of clinical follow-up.”

III the idea that this paper is a follow up to reference 8 is confusing given that the numbers are so different

Reviewer #2: (No Response)

7. PLOS authors have the option to publish the peer review history of their article (what does this mean?). If published, this will include your full peer review and any attached files.

Reviewer #1: No

Reviewer #2: No

---

## [Author Response · Author response to Decision Letter 2]

14 Apr 2021

Reviewer #1: I find this paper somewhat confusing still. The authors have tried to respond to my inquiries but as best as I can tell there is a barely statistically significant difference in groups that did and did not undergo angiography in the setting of symptomatic peripheral arterial disease.

This remains in my mind a not compelling and in fact a confusing story

I this remains confusing – there are three groups – two that had routine angiography one with and one without coronary artery disease and a third who did not undergo angiography. And the five year clinical follow up is that both groups who underwent coronary angiography were equivalent but had minor HR 0.347 p=0.049 reduction compared to those who did not. In other words regardless as to whether coronary artery disease was discovered and regardless as to whether an intervention was performed.

Is this correct?

If this is correct – what is the message that angiography is to be performed

Is it the case that there is benefit on the crossover group only or the entire group?

>>> We sincerely appreciate your comments. Your point is correct. Although the risk of MI in the CAG group (HR: 0.347, 95% CI: 0.121 to 0.998, p=0.049, Figure 1, F) was lower compared to the no CAG group, this is due to the no CAD group (HR: 0.204, 95% CI: 0.049 to 0.852). The risk of MI in the CAD group (HR: 0.523, 95% CI: 0.184 to 1.485) was lower than that of the group without CAG, but there was no statistical significance (This statistical values were not shwon on the table and figure). Therefore, this might be confusing to readers and reviewers. In the revised manuscript, all statements that the MI risk reduction due to the CAG test has been deleted and changed. Comparing 3 groups appears to be clinically important and relevant due to it is practically reflecting real world clinical practice. Some proportion of patients definitely refuse to undergo coronary angiography due to absence of symptoms, cost issues and other personal preferences. Thus we need to know the long term impact of the active coronary surveillance in advanced PAD patients undergoing PTA. 

II Moreover, the text is till confusing – take for example the next to last paragraph which reads

Third, all subjects in this study underwent CAG, and therefore our results are not generalizable to patients who do not receive CAG. Finally, all the PAD patients with significant CAD patients did not undergone PCI due to some reasons including clinical judgment based upon the discretion of the clinician and patient preference. Only 71.6% of all CAD patients underwent PCI or CABG before, after or at the same time as admission for PTA. This registry reflects the real-world practice of PAD patients.

1 the text is unclear – there are some extra words

2 was the benefit referred to in all patients who had CAG irrespective of whether they

a. had CAD

b. had an intervention

or were only the 71.6% of all CAD patients who underwent PCI or CABG considered?

>>> We sincerely appreciate your comments. We have made some changes to this statement in the limitation section as follows as “The results of our study were derived indirectly by comparison with the survival rates of other studies. In general, the 5-year survival rate of PAD patients is around 50% [15], and the survival rate of our study subjects, 88.5%, was very high. The no CAG group without cardiac function evaluation should be allocated to verify the effectiveness of routine CAG in PAD patients, but this is a problem that may be against research ethics. In this study, PTA was performed by an interventionist based on cardiology, the cardiac function of all subjects was evaluated non-invasively and/or invasively by the cardiologist. Therefore, the no CAG group in this study should not be confused with the patients that did not perform cardiac function evaluation.” Also, we have some changes in the method section as follows as "A total of 765 consecutive PAD patients underwent successful PTA and 674 patients underwent routine CAG. For the remaining 91 patients who did not receive the CAG, but the cardiac function was evaluated non-invasively by the cardiologist. "

>>>This latter issue is further confusing by the note on page “In the present study, the CAG group had reduced significantly the risk of myocardial infarction by 65.3% than the no-CAG group during 5 years of clinical follow-up.”

We sincerely appreciate your comments. Your point is correct. Although the risk of MI in the CAG group (HR: 0.347, 95% CI: 0.121 to 0.998) is lower compared to the no CAG group, this is due to the no CAD group (HR: 0.204, 95% CI: 0.049 to 0.852). The risk of MI in the CAD group (HR: 0.523, 95% CI: 0.184 to 0.1.485) was lower than that of the group without CAG, but there was no statistical significance. Therefore, in the revised manuscript, all statements that the MI risk reduction due to the CAG test has been deleted and changed. The main finding of this study is that routine CAG and subsequent PCI for significant CAD in symptomatic PAD patients undergoing PTA is safe and resulted in similar long-term survival as compared with the symptomatic PAD patients who undergoing PTA and who did not have CAD. Not surprisingly, repeat PCI was performed more frequently in PAD patients with CAD at long-term follow-up. We wanted to demonstrate the risk reduction of advanced PAD patients with significant CAD patients by providing long-term clinical benefit from active coronary revascularization as compared to those of PAD patients without significant CAD. Thus, wanted to show the clinical benefit of active concomitant coronary surveillance in advanced PAD patients undergoing PTA to improve their prognosis.

III the idea that this paper is a follow up to reference 8 is confusing given that the numbers are so different

>>> We sincerely appreciate your comments. We agree with your comments. This study differs from the study in Reference 8 in its study subjects and study duration. Therefore, the statement of reference 8 has been deleted and changed in the revised manuscript.

---

## [Decision Letter · Decision Letter 3]

29 Apr 2021

Long-term Outcomes of Peripheral Arterial Disease Patients with Significant Coronary Artery Disease undergoing Percutaneous Coronary Intervention

PONE-D-19-21312R3

Dear Dr Rha

We’re pleased to inform you that your manuscript has been judged scientifically suitable for publication and will be formally accepted for publication once it meets all outstanding technical requirements.

Kind regards,

Xianwu Cheng, M.D., Ph.D., FAHA

Academic Editor

PLOS ONE

Additional Editor Comments (optional):

Although the original reviewer#1 has still rejected, following third peer-review processes, the authors satisfactory addressed all of original coments.

Reviewers' comments:

Reviewer's Responses to Questions

**Comments to the Author**

1. If the authors have adequately addressed your comments raised in a previous round of review and you feel that this manuscript is now acceptable for publication, you may indicate that here to bypass the “Comments to the Author” section, enter your conflict of interest statement in the “Confidential to Editor” section, and submit your "Accept" recommendation.

Reviewer #2: All comments have been addressed

2. Is the manuscript technically sound, and do the data support the conclusions?

Reviewer #2: Yes

3. Has the statistical analysis been performed appropriately and rigorously? 

Reviewer #2: Yes

4. Have the authors made all data underlying the findings in their manuscript fully available?

Reviewer #2: Yes

5. Is the manuscript presented in an intelligible fashion and written in standard English?

Reviewer #2: Yes

6. Review Comments to the Author

Reviewer #2: (No Response)

7. PLOS authors have the option to publish the peer review history of their article (what does this mean?). If published, this will include your full peer review and any attached files.

Reviewer #2: No

---

## [Editor Report · Acceptance letter]

7 May 2021

PONE-D-19-21312R3 

Long-term Outcomes of Peripheral Arterial Disease Patients with Significant Coronary Artery Disease undergoing Percutaneous Coronary Intervention 

Dear Dr. Rha:

I'm pleased to inform you that your manuscript has been deemed suitable for publication in PLOS ONE. Congratulations! Your manuscript is now with our production department. 

Kind regards, 

on behalf of

Associate Prof. Xianwu Cheng 

Academic Editor

PLOS ONE